# Low Volume, Home-Based Weighted Step Exercise Training Can Improve Lower Limb Muscle Power and Functional Ability in Community-Dwelling Older Women

**DOI:** 10.3390/jcm8010041

**Published:** 2019-01-04

**Authors:** Jacqueline L. Mair, Giuseppe De Vito, Colin A. Boreham

**Affiliations:** 1School of Applied Sciences, Edinburgh Napier University, Sighthill Campus, Edinburgh EH11 4BN, UK; 2Institute for Sport and Health, University College Dublin, Bellfield, Dublin 4, Ireland; giuseppe.devito@ucd.ie (G.D.V.); colin.boreham@ucd.ie (C.A.B.)

**Keywords:** exercise, health, ageing, physical fitness

## Abstract

Stepping exercise can be used as a scalable form of high intensity exercise to enhance important aspects of physical fitness in older populations. The addition of supplementary weights increases the resistive element of stepping, with the potential for training improvements in muscular strength, power, and functional abilities alongside other fitness outcomes. The aim of this study was to evaluate the effects of a low-volume, home-based weighted step exercise programme on muscular strength, power, and functional ability in previously inactive community-dwelling older women. Eleven participants, aged between 65–74 years, independently completed a six-week individualised and progressive step exercise training programme wearing a weighted vest. Knee extensor strength, lower limb power output, and physical function using a battery of functional tests were measured at baseline, following a 6-week control period, and again following the 6-week training programme. Following training, lower limb power output improved by 10–11% (*p* < 0.05) and was accompanied by a corresponding 9% (*p* < 0.01) improvement in stair climb time and 10% (*p* < 0.01) improvement in normalised stair climbing power, highlighting the beneficial effects of weighted stepping for transferable improvements in functional fitness. The magnitude of observed training improvements suggest that weighted step training has the potential to prolong independence and prevent age-related health conditions such as sarcopenia.

## 1. Introduction

The degenerative loss of skeletal muscle mass and strength, known as sarcopenia [1], is one of the main physical problems encountered by the ageing population. Cross-sectional studies have shown that after the 5th decade of life, muscle strength decreases at a rate of approximately 12% to 15% per decade, with more pronounced declines after 65 years of age [2]. Muscle power declines at an even faster rate of 3–4% per annum between the ages of 65 and 89 years [3]. Unsurprisingly, such rapid declines in muscle strength and power are associated with an increasing difficulty in performing activities of daily living (ADL’s), such as climbing stairs, rising from a chair and undertaking household chores [4,5]. Women are generally weaker than men across all age categories [6] and may reach levels of strength that fall below the threshold required to accomplish typical ADL’s, by 70 years of age [3,7]. If such losses in strength and function are not addressed in a timely fashion, there is an increased likelihood of accelerated decline towards disability, dependency, and chronic conditions such as osteoporosis.

Progressive resistance training (PRT) programmes have consistently produced improvements in muscle function and functional ability in older adults [2,8], as well as a reduction in the symptoms of various chronic diseases such as arthritis, depression, type II diabetes, osteoporosis, sleep disorders, and heart disease [9]. Power training (involving higher velocity resisted movements) interventions have been shown to be more effective than traditional PRT (low velocity) for improving functional outcomes in older adults [10,11,12]. Evidence such as this led to the inclusion of muscle strengthening and balance and coordination recommendations for adults and older adults in global physical activity guidelines [13]. However, the proportion of adults participating in such activities remains low and shows a decline with increasing age [14,15]. Research has highlighted several barriers to participation in PRT programmes in older adults, including access to facilities and equipment, lack of time, cost, inconvenience, and a lack of age-appropriate classes [16]. Additionally, reasons for ceasing participation in resistance training include injury, illness, holidays, and issues with fitness centres or staff [17]. It is therefore necessary to consider alternative PRT opportunities that address these barriers and issues.

Previous research has shown dynamic high-velocity resistance training with supplementary loads to be effective in improving lower limb strength and power after 12 weeks in elderly adults [18,19]. Weighted stair climbing has also been shown to improve knee extensor strength, leg press power and stair climbing power after 12 weeks in older adults [20]. Stepping exercise is a multi-joint functional task similar to stair climbing [21] that may enhance important aspects of physical and functional fitness in an elderly population. Although previous research has shown that short bouts of stepping exercise improve aerobic fitness but not muscle strength in a middle-aged cohort [22], it is possible that stepping with additional supplementary loads will provide the necessary stimulus to effect changes in muscular strength and power, and consequently functional fitness, in older adults. Thus, the aim of this study was to evaluate the effects of a six-week weighted step exercise programme on parameters of muscular strength, power, and functional ability in previously physically inactive elderly women.

## 2. Experimental Section

A priori power calculations were carried out based on a previous study reporting adaptations in muscle power in older adults [18]. Based on a paired-sample t-test with a 5% significance level and 80% power and assuming a minimal “physiologically important” change due to training in peak power of 55.9 ± 50.1 W, a minimum of 11 participants were required. Local university ethical approval (LS-12-67-Mair-Boreham) was obtained prior to the commencement of this study. Participants were recruited through a local area information bulletin and newspaper advertisement. Applicants to the study were asked to complete a screening questionnaire that included a detailed Health History Questionnaire [23] and questions about previous and current exercise participation. All participants were required to be physically inactive, that is, not taking part in any regular, planned physical exercise on two or more occasions per week. A total of 41 enquiries were received and 22 participants completed the screening process. Of the latter, 15 were deemed eligible for the study and agreed to take part. Four withdrew from the study following initial baseline assessments (two due to an unrelated health condition, and two due to lack of availability), therefore a total of 11 healthy sedentary older females completed the study and were included in the analysis. Informed consent was obtained from all participants prior to commencement of the study and all agreed to maintain their normal daily routines and dietary intake during the study duration. On all occasions, participants were instructed to report to the laboratory, having consumed no caffeine or alcohol for at least 12 h, and no food for at least two hours prior to testing.

### 2.1. Study Design

Participants were required to complete their training independently and unsupervised in their own home. The study was of quasi-experimental wait-list cross-over design [24], used in previous research [25], which allowed all participants to complete a 6-week control period followed by a 6-week intervention. Testing sessions took place on weeks 0 (baseline), 6 (pre) and 13 (post).

### 2.2. Anthropometric and Resting Measurements

Height and weight were measured with participants lightly dressed and barefoot using a stadiometer (Holtain Ltd., Pembrokeshire, UK) to the nearest 0.1 cm and digital scales (Seca, Hamburg, Germany) to the nearest 0.1 kg, respectively. Resting heart rate (HR) and blood pressure (BP) were measured using a digital blood pressure monitor (A&D Company Ltd. Model UA-767, Tokyo, Japan). Participants rested in a seated position in a quiet, temperature-controlled room for a period of five minutes, after which measurements were performed twice on the left arm.

### 2.3. Functional Tests

#### 2.3.1. Stair Climb

Stair climb time, and the subsequent calculation of stair climb power (SCP), were used as a measure of functional ability. Participants were asked to safely ascend a standard, well-lit, public staircase consisting of 10 steps (riser height: 20 cm), as quickly as possible. They were instructed to use the handrail only if necessary and then only lightly, and to begin after a short countdown (“three, two, one, GO”). Timing commenced on the command “GO” and stopped when the participant’s foot reached the top step. Time was recorded to the nearest 0.01 s, and the best performance from three trials was recorded.

Stair climbing power (SCP) was calculated using the formula:*P* = *F* × *v*(1)
*F* = (*m* × *a*)(2)
*v* = (*d*/*t*)(3)
where *P* is power, *F* is force, *v* is velocity, *m* is body mass in kilograms, *a* is the acceleration due to gravity (9.81), *d* is distance in meters and *t* is time in seconds.

#### 2.3.2. Chair Rise

The five-repetition chair rise was used as an indicator of lower-limb muscle power in older adults. The participant was instructed to rise from a chair of height 40.5 cm from the ground, five times consecutively, and as quickly as possible. Arms were crossed in front of the chest throughout the test. Time was measured to the nearest 0.01 s using a stopwatch and the best performance from three trials was recorded.

#### 2.3.3. Four-Meter Walk

The 4 m walk test was used as an indicator of habitual walking velocity. Participants were instructed to walk a distance of four meters at their usual pace. The course was marked with tape and participants were timed from the instant they began the test to the point at which the foot fully crossed the four meter mark. Time was measured to the nearest 0.01 s using a stopwatch, and the best performance from three trials was recorded.

### 2.4. Lower Limb Muscle Strength

Participants were tested for isometric maximal voluntary contraction (MVC) of the knee extensor muscle group on the dominant lower limb using a dynamometer (Cybex NORM, Lumex, Inc., Ronkonkoma, NY, USA) with a sampling rate of 100 Hz. Participants were seated comfortably in the dynamometer chair and firmly strapped to the seat with the hips and knees flexed at 100° and 90°, respectively. The lateral femoral epicondyle of the dominant limb was aligned with the rotational axis of the dynamometer and an ankle cuff secured the lower leg to the machine lever arm, 2 cm superior to the lateral malleolus. The individual positioning of each participant, of the seat, backrest, dynamometer head and lever arm length were recorded and replicated on subsequent testing sessions. A standard warm-up and familiarisation involving submaximal and maximal contractions was performed prior to testing.

The MVC task consisted of a rapid increase to a maximum knee extensor contraction. Participants were carefully instructed to contract the knee extensors “as quickly and forcefully as possible”. They were verbally encouraged to achieve a maximum and to maintain it for at least 3 s before relaxing. Visual feedback was provided on a computer screen allowing participants to follow their performance. MVC was calculated as the largest 50 ms average reached within any single force recording. A minimum of three attempts were performed, each separated by a two-minute rest period, and the trial resulting in the highest force value was chosen as MVC.

### 2.5. Lower Limb Power

Lower limb power output was assessed by means of a modified Wingate anaerobic test (WAnT) [26]. Since peak power output usually occurs within the initial 5 s of the test [27] and this was our variable of interest, a shortened version of the original 30 s WAnT was deemed appropriate for this population. Participants performed a warm-up and familiarisation session prior to testing on every occasion which included a three-minute warm-up at 60 revs/min followed by two 5 s sprints and one 8 s sprint from a rolling start of 70 revs/min. A two-minute rest period was permitted between sprints. Following this, the full WAnT protocol was performed which consisted of three 8 s sprints from a rolling start of 70 revs/min interspersed with 2 min of active recovery and 4 min of passive recovery. Each sprint was preceded by a two-minute warm up at 40 W and 60 revs/min. The pedal rate was increased to 70 revs/min and following a countdown of “three, two, one, GO!”, participants performed the WAnT style sprint. Participants then cycled freely without load to allow a short recovery, before resting for four minutes. Peak power output was recorded from the first 5 s of test data and mean power from the full 8 s. The average of three trials was used in analysis. 

### 2.6. Training Programme

The training programme consisted of 54 exercise sessions, which were divided into three training sessions/day on three days/week over a six-week period. Participants were allowed the freedom to choose the days and times of each session. All training sessions were carried out in the participant’s own home, using an adjustable portable step (Reebok International Ltd., Canton, MA, USA) and a weighted vest (Centurion Rugby, Dewsbury, UK). The training protocol included four sets of 10 step-up repetitions performed as quickly as possible, two sets of which were executed with the right leg leading and two with the left. The step height remained at 20 cm throughout the six-week training programme, but intensity was progressively increased through supplementary loading increments. This was prescribed to the participant in relation to their body mass (Table 1) and in line with previous research [22]. They were required to maintain a training diary over the course of the programme, recording the date, time and Rating of Perceived Exertion (RPE) score [28] and including any notes relating to the performance of each session. Weekly telephone and email contact were maintained throughout the training period to ensure the participants were progressing without problems.

Total work done during each week of the training programme was calculated using the following formula:*W* = *F* × *d*(4)
*F* = ((*m* + *l*) × *a*)(5)
where *W* is work done, *F* is force, *d* is distance in meters, *m* is body mass in kilograms, *l* is supplementary load in kilograms, and *a* is the acceleration due to gravity (9.81). Distance was calculated as the total number of step ups performed (positive work) multiplied by the height of the step (20 cm).

### 2.7. Statistical Analysis

All analyses were performed using IBM SPSS Statistics version 23 (IBM Corporation, Armonk, NY, USA). The percentage change (defined as the difference between baseline and post-training test values, divided by the baseline value) was calculated to provide an estimate of the magnitude of change for each variable. A one-way repeated measures analysis of variance (ANOVA) with post-hoc analysis using Bonferroni correction for multiple comparisons was used to test for difference in dependent variables across the three testing sessions. Mauchly’s test of sphericity was also consulted and where significant, the Greenhouse-Geisser statistic was used to check for differences. Significance was accepted as *p* < 0.05. Effect size values were calculated using partial eta squared (*η*_p_^2^) whereby 0.01 is considered a small effect, 0.06 a medium effect, and 0.14 a large effect [29].

## 3. Results

Adherence to the exercise programme, based on training diary information, was 97%. Training volume, represented by the total work done while ascending the step (excluding contribution from negative work during descent), increased from 47.8 kJ in week one to 52.5 kJ in week six. RPE scores also increased from 10.4 (light) in week one to 13.1 (somewhat hard) in week six (Table 1).

Table 2 presents the key study variables. Body mass, resting HR and resting BP were unchanged as a result of the six-week training programme. Analysis of the modified WAnT revealed a significant improvement in absolute peak power (mean change pre-post = 62.6 W, 10.2%, *p* = 0.037) and normalised peak power (0.99 W·kg^−1^, 10.9%, *p* = 0.036) following the training programme. However, absolute and normalised mean power output did not show a significant change with training (mean change pre-post = 24.4 W, 8.4%, *p* = 0.238; 0.35 W·kg^−1^, 7.9%, *p* = 0.266, respectively). Similarly, there was no significant change in quadriceps muscle strength (mean change pre-post = 9.03 Nm, 9.2%, *p* = 0.350). Of the functional tests, stair climb time (mean change pre-post = 0.45 s, −9.4%, *p* = 0.002) and normalised stair climb power (0.44 W·kg^−1^, 10.6%, *p* = 0.004) were significantly improved following training. Absolute stair climb power (mean change pre-post = 40.15 W, 15%, *p* = 0.082), 4-m walk time (0.12 s, −3.7%, *p* = 0.314) and chair rise time (0.67 s, −6.5%, *p* = 0.118) were unchanged as a result of the six-week training programme.

## 4. Discussion

Older adults currently represent the fastest growing population segment worldwide [30]. Women live longer than men but spend a greater number of years before death being disabled [31,32]. Indeed, rates of falling are higher in older women than in older men and continue to increase with age above 65 years [33]. It has also been reported that women’s functional fitness is more likely than men to decline to a level which cannot sustain independent living [3]. It is therefore of paramount importance to establish effective exercise training interventions that can improve health and prolong independence in ageing females.

The main finding of this study is that muscular power and certain aspects of functional ability can be significantly improved in older female adults following six weeks of an extremely low volume weighted step exercise programme performed at home. The training, which was designed to reflect a high velocity PRT programme, was individually prescribed to each participant in a manner that allowed easy integration into the normal daily routine, reflected by a 97% adherence rate. Overall, the home-based programme was time-efficient, low-cost, and well tolerated by the older female participants. The programme may therefore be a viable intervention strategy for older adults who are not meeting guidelines for muscle strengthening activity.

The results show an improvement in absolute and normalised lower limb peak power of between 10% and 11%, respectively, indicating a clinically meaningful change [34]. These findings are supported by a corresponding 9% improvement in stair climb time and 10% improvement in normalised stair climbing power, highlighting the beneficial effects of weighted stepping for transferable improvements in the performance of ADL’s. Although different training protocols and testing procedures prevent a direct comparison, the magnitude of muscular power improvements are in line with previous weighted training programmes [18,35,36]. These improvements were observed despite a greatly reduced training volume (in terms of work done, time commitment, external loading, and training programme duration) compared with these other studies [18,35,36], providing further insight into the minimum dose necessary to effect changes in muscle power.

It should be noted that, while significant improvements in peak power output were observed, mean power output changes were not evident. Despite multiple familiarisation trials within the protocol, post-hoc comparisons for absolute and normalised mean power output suggest that results were partially confounded by a learning effect. The extreme nature and high level of motivation demanded by the modified WAnT protocol suggests a longer familiarisation period for untrained older participants who are unaccustomed to cycling may be necessary. Previously, Miszko and colleagues [5] failed to show any clear improvement in parameters of power obtained from a 30 s WAnT, following a 16-week power training protocol in older adults. This may have been due to the longer WAnT protocol used, since improvements in physical function were evident as a result of training. Conversely, Shaw and Snow [36] reported a 13% increase in maximum leg power using as shorter duration modified 15 s WAnT and following a longer nine-month multi-task intervention (including stepping, squats, chair rises, lunges and toe raises) with weighted vests. The testing method and protocol used to detect changes in lower limb power output appears to be of importance, and further research may be needed to elucidate what protocol is most suited to older and inactive populations. Certainly, when using methods such as a modified WAnT, multiple familiarisation sessions may be necessary prior to testing intervention effects.

Knee extensor muscle strength showed a modest but non-significant increase of 9% following training. This change in lower limb muscle strength is lower than the 16–20% improvements previously reported following weighted exercise programmes lasting 12–36 weeks [35,36], but is likely reflective of the training focus on high velocity and low load, the testing protocol adopted, the shorter training duration and the extremely low training volume (48–53 kJ per week) compared with these other studies. For example, Shaw and Snow [36] used isokinetic dynamometry to test for changes in knee extensor muscle strength, while Bean and colleagues [35] used a seated double leg press protocol on pneumatic strength testing equipment. Both of these methods test dynamic strength, whereas the current study adopted an isometric testing protocol. While maximal isometric testing of the knee extensors is commonly used to test muscle strength and is indeed recommended for the assessment of musculoskeletal ageing [37], in this instance it was not specific to the training modality and is likely to have limited the ability to detect change. In addition, the study was powered to detect changes in lower limb power but not other outcomes of interest.

Functional test scores showed statistically significant improvements for stair climb time and the subsequent calculation for normalised stair climb power. The 9–10% improvement in physical function identified by these tests is suggestive of a substantial clinically meaningful change [38]. The lack of change in chair rise time and habitual walking velocity may be attributed to the specificity of the training programme, as highlighted in previous research [18,20]. In addition, although physically inactive, the participants were healthy non-frail community-dwelling older adults and therefore it is likely that their level of function at baseline was already greater than the minimum functional threshold. For example, mean baseline values for the five-repetition chair rise of 10.7 s are reflective of an above average performance, based on threshold normative values of 11.4 s for 60–69 year olds [39]. Calculating mean habitual gait velocity (1.21 m/s) from 4 m walk time revealed that, at baseline, participants were also of better than average predicted life expectancy. Predicted life expectancy at the median for age and sex has been reported to occur at approximately 0.8 m/s, with faster gait speeds predicting life expectancy beyond the median [40]. Thus, despite improvements in muscle power, corresponding improvements in function, as measured by chair rise and habitual walking velocity, may have been more difficult to detect.

An encouraging result of the study was the high adherence to the programme. Maintaining long term participation in traditional strength and conditioning programmes has proved problematic in older cohorts and in particular women [41]. Therefore, more manageable, time-efficient, and convenient interventions such as weighted stepping might help resolve this issue. Furthermore, the training was cost effective, discreet and could easily be carried out in the home using a step (e.g., bottom step of a staircase) and some household items in a backpack. As such, there is potential for such a programme to be implemented on a larger scale, which may lead to significant impact in terms of prolonged independence in older adults and a significant cost saving for health services.

While this particular study examined the effects of weighted stepping on muscular strength power, and functional ability, previous research has shown stepping exercise to have beneficial effects on aerobic fitness [22]. It is possible that an adapted programme of weighted stepping may confer simultaneous benefits in aerobic and muscular fitness. Loy and colleagues [42] reported the combined cardiorespiratory and musculoskeletal benefits resulting from stair climbing exercise, with concurrent improvements in aerobic capacity (10–11%) and quadriceps muscle strength (10.5%) after 12 weeks of training in previously sedentary middle-aged females. Furthermore, Boreham et al. [43] have shown that stair climbing leads to wider cardiovascular health benefits such as improved aerobic fitness and increased HDL cholesterol levels. An activity that provides concurrent improvement in muscle strength and power as well as cardiorespiratory fitness and health benefits may be a more desirable exercise intervention for those failing to meet the recommended levels of physical activity (which include aerobic, strength and balance for older adults). It may also make physical activity guidelines more achievable for older adults. Further research over a longer training period and possibly with a higher training volume in terms of energy expenditure, is required to elucidate the combined effects of weighted stepping exercise on aerobic and other aspects of functional fitness.

There are some limitations to this study that should be considered. The study sample was restricted to healthy Caucasian females within a specific age range and therefore the reported results may not be truly representative of the general population. It is suspected that many of the non-significant differences between the pre and post-intervention measures may have been related to the small sample size. Recruitment was based on an a-priori power calculation for changes in the outcome measure considered to be of the most importance. As such, the sample size may not have resulted in sufficient power to detect changes in other parameters of interest. Moreover, due to a small number of dropouts the reported results may be slightly under powered. A learning effect was also evident for some tests, which may have been due to insufficient familiarisation. Furthermore, some of the results observed following the training programme might have been limited due to the restrictions imposed by time constraints, and the practicality of the study protocols in relation to the populations of interest. Longer training periods would likely enhance observed benefits, possibly revealing more significant findings, and thus strengthening the conclusions made.

## 5. Conclusions

The results of this study show, for the first time, that low-volume stepping exercise with an external load can significantly improve lower limb muscle power and functional ability in older females by approximately 10%. As ageing is associated with declines in muscle strength and power by 12–15% per decade after 50 years of age [2,3], the observed training improvements observed as a result of this weighted stepping intervention could potentially prolong independence by almost 10 years in older women. As stated by Sayers [44], there is a critical need for resistance training protocols to appeal to older adults and promote long-term participation, and weighted stepping has the potential to fulfil these criteria.

## Figures and Tables

**Table 1 jcm-08-00041-t001:** Progressive training model, including relative prescribed external loads across the six-week programme and the average absolute external load for each week.

Week	Prescribed Load (% Body Mass)	Average External Load (kg)	Total Work Done (kJ)	Average RPE
1	zero	zero	47.8 ± 7.7	10.4 ± 0.8
2	5	3.32 ± 0.51	50.1 ± 8.0	11.5 ± 0.5
3	5	3.42 ± 0.52	50.2 ± 8.0	11.4 ± 0.8
4	7.5	5.16 ± 0.85	51.4 ± 8.3	12.3 ± 0.6
5	10	6.67 ± 1.04	52.5 ± 8.4	13.3 ± 0.5
6	10	6.75 ± 1.02	52.5 ± 8.4	13.1 ± 0.4

Values are means ± SD; Rating of Perceived Exertion (RPE).

**Table 2 jcm-08-00041-t002:** Main outcome variables at baseline, pre-intervention (pre) and post-intervention (post).

	Baseline	Pre	Post	Change (%)		Effect Size
				Baseline-Pre	Pre–Post	*η* _p_ ^2^
Age (years)	67.4 ± 3.53					
Stature (m)	1.59 ± 0.05					
Body Mass (kg)	67.6 ± 10.9	67.7 ± 10.3	67.4 ± 9.9	0.1	−0.4	0.015
Resting HR (bpm)	80 ± 14	76 ± 14	75 ± 15	−5.0	−1.3	0.139
Systolic BP (mmHg)	141 ± 23	132 ± 25	125 ± 16	−6.2	−5.3	0.379
Diastolic BP (mmHg)	82 ± 13	80 ± 13	80 ± 12	−4.2	0.4	0.054
Stair Climb Time (s)	5.11 ± 0.76	4.79 ± 0.65	4.34 ± 0.61 **	−6.3	−9.4	0.556
Stair Climb Power (W)	262.5 ± 44.4	267.4 ± 47.0	307.6 ± 44.3	1.9	15.0	0.381
Stair Climb Power (W/kg)	3.92 ± 0.55	4.17 ± 0.57	4.61 ± 0.67 **	6.5	10.6	0.573
4-m walk (s)	3.14 ± 0.40	3.26 ± 0.4	3.14 ± 0.3	3.8	−3.7	0.033
Gait velocity (m/s)	1.21 ± 0.12	1.19 ± 0.12	1.24 ± 0.10	−1.7	4.2	0.165
Chair rise (s)	10.70 ± 1.62	10.36 ± 1.06	9.69 ± 0.83	−3.2	−6.5	0.167
MVC (Nm)	96.3 ± 11.9	98.1 ± 18.5	107.1 ± 17.3	1.9	9.2	0.231
Peak Power (W)	582.1 ± 117.0	612.9 ± 135.2	675.5 ± 153.4 *	5.3	10.2	0.566
Peak Power (W/kg)	8.7 ± 1.3	9.1 ± 1.2	10.1 ± 1.8 *	4.6	11.0	0.550
Mean Power (W)	255.3 ± 63.5	290.7 ± 65.5	315.1 ± 79.7	13.9	8.4	0.526
Mean Power (W/kg)	3.9 ± 1.1	4.4 ± 1.0	4.7 ± 1.1	12.8	6.8	0.532

Mean ± SD; Heart Rate (HR), Blood Pressure (BP), Maximal Voluntary Contraction (MVC); Post-hoc comparisons performed with Bonferroni adjustment; * denotes significant difference between pre and post measures at *p* < 0.05; ** denotes significant difference between pre and post measures at *p* < 0.01.

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
