# Peer review of "Low Volume, Home-Based Weighted Step Exercise Training Can Improve Lower Limb Muscle Power and Functional Ability in Community-Dwelling Older Women"

_jcm, 2019, doi:10.3390/jcm8010041_

Round 1
Reviewer 1 Report
The manuscript by Mair et al. analyzed the effects of low-volume, home-based weighted step exercise program on muscular strength and function in older women. This manuscript was well-written, and well-considered. Some point should be revised to enhance its scientific impact.
Major point
In abstract, the authors described that twelve participants competed the study (line 19). However, in the Experimental Section, the authors described that “a total of 11 healthy sedentary older females were included” (line 85). The authors should make a flow diagram or participants enrollment, allocation, follow-up and analysis, and should describe how many participants were analyzed in this study.
The authors performed six weeks of “an extremely low volume” weighted step exercise program in this study (line 225). How did the authors define “extremely low volume” used in this study? Although average load (was this the weight of the vest?) was described in Table 1, the authors should show the reason why this load was “an extremely low volume”. The comparison with previously performed study is required. This point is important because it would be easily assumed that repeated exercise causes muscle hypertrophy.
According to CONSORT statement, the authors should describe the limitation of this study. Although the required sample size was theoretically estimated (line 76), the small sample size might be one of the limitations of this study. Furthermore, because some participants dropped out this study, the authors performed per protocol set analysis. This point might weaken the quality of this study.
Minor point
The authors should describe the approval number of this study (line 76).
The authors should describe short conclusion in the Conclusion section. It seemed to the introduction of the Discussion.
Author Response
In abstract, the authors described that twelve participants competed the study (line 19). However, in the Experimental Section, the authors described that “a total of 11 healthy sedentary older females were included” (line 85). The authors should make a flow diagram or participants enrollment, allocation, follow-up and analysis, and should describe how many participants were analyzed in this study.
Thank you for highlighting this error in the abstract. This has been amended. We have also amended the experimental section slightly for enhanced clarity regarding recruitment and drop out (specifying all dropouts occurred following baseline assessment). Due to the study design we did not have allocation to groups. We believe the section is now improved and clearly describes recruitment, enrolment, follow-up and analysis therefore it is our view that a flow diagram is not necessary.
The authors performed six weeks of “an extremely low volume” weighted step exercise program in this study (line 225). How did the authors define “extremely low volume” used in this study? Although average load (was this the weight of the vest?) was described in Table 1, the authors should show the reason why this load was “an extremely low volume”. The comparison with previously performed study is required. This point is important because it would be easily assumed that repeated exercise causes muscle hypertrophy.
Training volume is defined here in terms of energy expenditure during the exercise bouts, number of repetitions, and training programme duration. We explain that the training volume (work done while ascending the step) equated to only 43-53 kj of energy expenditure per week. Although we did not monitor the time taken to complete the total number of training bouts, from pilot testing, 4 sets of 10 step repetitions performed as quickly as possible took approximately 1 to 2 minutes to complete depending on rest intervals between bouts. This is much lower than traditional dynamic resistance training programmes which tend to involve higher loads and a greater number of sets than was used in this study.
Much of the literature on high intensity interval training refers to the exercise programme as being low volume in terms of energy expenditure and time commitment (Gibala et al., 2012). Within resistance training, low volume is generally defined as involving fewer set and repetitions with higher loads (Schoenfeld and Grgic, 2018). We believe the step exercise programme implemented in this study reflects both of these definitions, but due to the very low external loading (average 6.75 kg in week 6), it is extremely low in comparison to usual functional or resistance training.
We have added at line 247 what aspects of the training we believe to be lower volume and have reiterated the studies that we are comparing to by including the references again. At line 314 we have also added that aerobic fitness adaptations will possible require higher volume of training in light of the low energy expenditure associated with the current training programme.
Gibala MJ, Little JP, MacDonald M and Hawley JA (2012). Physiological adaptations to low-volume, high-intensity interval training in health and disease. J Physiol, 590(Pt 5): 1077-1084.
Schoenfeld B and Grgic J (2018) Evidence-Based Guidelines for Resistance Training Volume to Maximize Muscle Hypertrophy. J Strength Cond Res, 40(4): 107-112.
According to CONSORT statement, the authors should describe the limitation of this study. Although the required sample size was theoretically estimated (line 76), the small sample size might be one of the limitations of this study. Furthermore, because some participants dropped out this study, the authors performed per protocol set analysis. This point might weaken the quality of this study.
We have included a limitations section where we address the sample size.
Minor point
The authors should describe the approval number of this study (line 76).
The reference provided by the ethics committee is now included at line 77.
The authors should describe short conclusion in the Conclusion section. It seemed to the introduction of the Discussion.
The conclusion has been revised and shortened.
Reviewer 2 Report
This is an interesting report confirming previous works on climbing stairs as the preventive measure to slow the progress of sarcopenia observed in aging people. This study was addressed to female individuals in modern society as more likely to become disabled after 65 years of age due to a longer lifespan in comparison to males. The experimental protocol, study course, results and discussion sections are written in a clear and concise manner. The only issues raised by the reviewer during reading this report are the novelty and the number (n=11) of individuals chosen for study which are doubtful.
Author Response
Respectfully, we believe this study is the first to explore strength, power, and functional ability adaptations to a low volume weighted stepping intervention, performed at home and unsupervised in a community dwelling older female sample. Previous research using external loading during exercise has focused on stair climbing or multifunctional movements but we are not aware of any other research focusing on stepping exercise. We believe the study describes a novel approach to home-based exercise that has the potential to improve several aspects of physical function and warrants further exploration. The study was slightly under-powered based on a priori calculations and we have now highlighted this within a new limitations section within the paper.
Reviewer 3 Report
Broad Comments: Overall a good study, and well put together. My major concern is the use of percent change scores for analysis. I do not agree with that method of statistical analysis. Percent change scores are a fine descriptive statistic but should not be utilized as an inferential statistic. Was an ANCOVA utilizing the baseline scores as the covariate attempted? As it is, this is a good experiment and reported how the percent change from this study compares to those of previous studies can provide good and valid information to further the field, but as this is written, I cannot agree to it.
Specific Comments
Line 18 comma after “power” (I know it seems a bit nitpicky, but with so many combinations in additions to lists in your paper the oxford comma helps separate them)
Line 22 add comma after the word “period” and before “and again”
Lines 38 and 41 Make sure font is consistent throughout
Line 42 add comma after the word “dependency”
Line 55 have “as well as” start a new sentence
Line 56 comma after “holidays”
Line 70, add comma after “power”
Methods sections:
2.3 Functional Tests (line 103) Sentence is unnecessary—just utilize the subsequent subheadings
Results section:
Line 203 Move last part of sentence “indicating the participants found the training slightly more difficult as the load was increased over the six-week (line 204) period” to discussions section—it speaks too much to interpretation
Discussions section:
Line 229 Add comma in after “low-cost” and before “and”
Delete gap (line 257)
Delete “indeed” in line 278
Line 321 needs a period at the end of the sentence
Author Response
Broad Comments: Overall a good study, and well put together. My major concern is the use of percent change scores for analysis. I do not agree with that method of statistical analysis. Percent change scores are a fine descriptive statistic but should not be utilized as an inferential statistic. Was an ANCOVA utilizing the baseline scores as the covariate attempted? As it is, this is a good experiment and reported how the percent change from this study compares to those of previous studies can provide good and valid information to further the field, but as this is written, I cannot agree to it.
Percent change scores are presented alongside results from inferential statistical analysis to assist with understanding the magnitude of change and for comparison with other reported literature. We did not perform statistical analysis on percentage values. We have moved the * symbol (indicating significant differences) in table 2 to avoid any confusion and have specified in the legend that the * symbol relates to the comparison between pre- and post-intervention values.
A repeated measures ANOVA was deemed appropriate due to the nature of the study design (1 group tested at three separate time points). When discussing the effect of the training programme we have compared pre-test values with post-test values rather than the baseline or the average (pre-test - baseline/2) as a more conservative means of estimating effect. If we were to use one of the aforementioned methods, we would be reporting larger effects. We also use the Bonferroni post-hoc test to account for multiple comparisons and to identify where differences occurred. Between baseline and pre-test (i.e. the control period) we did not find any significant differences, while between pre-test and post-test we identified some significant effects.
Specific Comments
Line 18 comma after “power” (I know it seems a bit nitpicky, but with so many combinations in additions to lists in your paper the oxford comma helps separate them)
addressed
Line 22 add comma after the word “period” and before “and again”
addressed
Lines 38 and 41 Make sure font is consistent throughout – we have used the journal template/styles to format the paper and are unable to see a difference in font at these lines. We presume issues such as font changes will be addressed at proof stage by the editorial office.
Line 42 add comma after the word “dependency”
Line 55 have “as well as” start a new sentence
Line 56 comma after “holidays”
Line 70, add comma after “power”
All addressed
Methods sections:
2.3 Functional Tests (line 103) Sentence is unnecessary—just utilize the subsequent subheadings
addressed
Results section:
Line 203 Move last part of sentence “indicating the participants found the training slightly more difficult as the load was increased over the six-week (line 204) period” to discussions section—it speaks too much to interpretation
addressed
Discussions section:
Line 229 Add comma in after “low-cost” and before “and”
addressed
Delete gap (line 257)
addressed
Delete “indeed” in line 278 – substituted for ‘for example’ as the sentence supports the point made in the previous one.
Line 321 needs a period at the end of the sentence
addressed
Round 2
Reviewer 1 Report
The authors fully answered my questions.